# Innate Immune Modulation by GM-CSF and IL-3 in Health and Disease

**DOI:** 10.3390/ijms20040834

**Published:** 2019-02-15

**Authors:** Francesco Borriello, Maria Rosaria Galdiero, Gilda Varricchi, Stefania Loffredo, Giuseppe Spadaro, Gianni Marone

**Affiliations:** 1Division of Gastroenterology, Boston Children’s Hospital and Harvard Medical School, Boston, MA 02115, USA; 2Department of Translational Medical Sciences and Center for Basic and Clinical Immunology Research (CISI), University of Naples Federico II, 80131 Naples, Italy; mrgaldiero@libero.it (M.R.G.); gildanet@gmail.com (G.V.); stefanialoffredo@hotmail.com (S.L.); spadaro@unina.it (G.S.); 3WAO Center of Excellence, 80131 Naples, Italy; 4Institute of Experimental Endocrinology and Oncology “Gaetano Salvatore” (IEOS), National Research Council (CNR), 80131 Naples, Italy

**Keywords:** GM-CSF, IL-3, innate immunity, sepsis, experimental autoimmune encephalomyelitis, rheumatoid arthritis, allergy, atherosclerosis, inflammatory bowel diseases, cancer immunotherapy, trained immunity

## Abstract

Granulocyte-macrophage colony-stimulating factor (GM-CSF) and inteleukin-3 (IL-3) have long been known as mediators of emergency myelopoiesis, but recent evidence has highlighted their critical role in modulating innate immune effector functions in mice and humans. This new wealth of knowledge has uncovered novel aspects of the pathogenesis of a range of disorders, including infectious, neoplastic, autoimmune, allergic and cardiovascular diseases. Consequently, GM-CSF and IL-3 are now being investigated as therapeutic targets for some of these disorders, and some phase I/II clinical trials are already showing promising results. There is also pre-clinical and clinical evidence that GM-CSF can be an effective immunostimulatory agent when being combined with anti-cytotoxic T lymphocyte-associated protein 4 (anti-CTLA-4) in patients with metastatic melanoma as well as in novel cancer immunotherapy approaches. Finally, GM-CSF and to a lesser extent IL-3 play a critical role in experimental models of trained immunity by acting not only on bone marrow precursors but also directly on mature myeloid cells. Altogether, characterizing GM-CSF and IL-3 as central mediators of innate immune activation is poised to open new therapeutic avenues for several immune-mediated disorders and define their potential in the context of immunotherapies.

## 1. Introduction

Granulocyte-macrophage colony-stimulating factor (GM-CSF) and inteleukin-3 (IL-3) are hematopoietic factors that belong to the β common ([βc]/CD131) family of cytokines together with IL-5 and KK34 (the latter has been identified in some species but its ortholog is a pseudogene in humans and mice) [1]. While the effects of IL-5 are mainly restricted to eosinophils [2,3], GM-CSF and IL-3 exert a broader role on the modulation of the innate immune response by engaging heterodimeric receptors composed of a shared βc subunit and a specific α subunit (CD116 for GM-CSF, CD123 for IL-3) that signal through the JAK2-STAT5 pathway as well as other kinases [1]. Their role in steady-state hematopoiesis is dispensable (except for the development/maintenance of alveolar macrophages of which absence leads to pulmonary alveolar proteinosis, non-lymphoid tissue dendritic cells (DCs) and functional maturation of natural killer T cells) [4,5,6,7,8,9,10]. GM-CSF and IL-3 promote emergency myelopoiesis (i.e., generation and differentiation of myeloid cells upon infection or in the context of inflammation) [9,11,12] and also modulate effector functions of several innate immune cell subsets in vitro and in vivo [1,13,14,15,16,17,18,19,20,21]. Of note, preliminary results of clinical trials have shown that blocking antibodies against GM-CSF and IL-3-specific receptor α subunits or βc subunits can be safe and effective in several autoimmune or inflammatory diseases [1,22]. This wealth of evidence has spurred renewed interest in evaluating the immunomodulatory potential of GM-CSF and IL-3 on the innate immune system and its therapeutic implication. In this review, we will summarize the biology of GM-CSF and IL-3 focusing on their role in the modulation of innate immune effector functions, the mechanistic relevance of GM-CSF and IL-3 in the development and progression of infectious, autoimmune and inflammatory disorders and how this new wave of knowledge is paving the way for novel therapeutic approaches (Figure 1 and Table 1).

## 2. Biology of GM-CSF and IL-3

GM-CSF and IL-3 are produced by several hematopoietic (e.g., lymphocytes) and non-hematopoietic (e.g., epithelial cells, and fibroblasts) cell subsets [1]. In particular, the cellular source can vary depending on the inducing stimulus and the experimental model. For example, airway epithelial cells challenged with allergenic stimuli produce GM-CSF, which in turn leads to allergic sensitization [23,24,25]. In human and murine models of multiple sclerosis (experimental autoimmune encephalomyelitis, EAE), GM-CSF is mainly produced by polarized T cells and memory B lymphocytes [16,26,27,28,29,30,31]. Although retinoic acid receptor (RAR)-related orphan receptor (ROR) γt-expressing T cells (e.g., Th17 lymphocytes) can produce GM-CSF [26,27], some groups have reported the STAT5-dependent expression of this cytokine by a specialized subset of T cell [29,30,31]. In addition, T cell expression of GM-CSF and IL-3 is specifically regulated by the transcription factor Bhlhe40 of which deletion does not reduce IFNγ or IL-17 production but impairs the induction of EAE [32,33]. GM-CSF is also expressed by T cells upon stimulation with IL-23 and plays a critical role in experimental models of T cell-dependent colitis [18,34]. IL-3 is produced by T cells upon activation and in particular under Th2 polarizing conditions such as allergic inflammation and parasitic infestation [35,36,37]. For example, thymic stromal lymphopoietin (TSLP)-activated DCs induce T cell production of IL-3, which plays a crucial role in basophil recruitment [35]. Another important source of GM-CSF and IL-3 is represented by innate immune cells. Type 3 innate lymphoid cells (ILC3s) are a major source of GM-CSF in the gut where it is required to modulate mononuclear phagocyte effector functions and induce the differentiation of regulatory T cells in the steady-state [38]. On the contrary, IL-23-induced GM-CSF production by ILC3s plays a critical role in the pathogenesis of ILC-driven colitis [39]. Innate response activator (IRA) B cells are a subset of B1a-derived B cells that produces GM-CSF and IL-3 in infection and atherosclerosis models, thereby modulating myeloid cell differentiation and maturation [11,40,41,42]. Finally, basophils are both the target and the source of IL-3, which also act in an autocrine manner to modulate basophil survival and cytokine production [19,20].

Beyond their contribution to emergency myelopoiesis, GM-CSF and IL-3 exert a prominent role in host defense and inflammation by modulating effector functions of mature myeloid cells. Cell specificity depends on the expression of specific receptor subunits: both GM-CSF and IL-3 can modulate monocytes, macrophages and eosinophils; GM-CSF also acts on neutrophils, while IL-3 is a potent regulator of basophil, mast cell and plasmacytoid DC activities [1,13,14,15,16,17,18,19,20,21]. The target cell subset also depends on the experimental model investigated, but in general GM-CSF and IL-3 contribute to ongoing inflammation by promoting cell survival, proliferation or activation. For example, GM-CSF unlocks a pro-inflammatory program in monocytes and their progeny and therefore plays an essential role in the pathogenesis of EAE [16]. Of note, GM-CSF and IL-3 also activate human monocytes. In combination with IL-4, they promote monocytes differentiation into DCs with Th2 and Th1 polarizing properties, respectively, for IL-3 and GM-CSF [43,44]. In addition, they promote the expression of CCL17 alone or in combination with IL-4 and enhance LPS-induced TNF-α production in in vitro models of trained immunity [13,14,15,45,46]. It is also worth mentioning that, in some experimental models, GM-CSF has shown an anti-inflammatory effect [17,38,47,48], highlighting the context-specific role of GM-CSF and IL-3 in inflammation.

IL-3 specific receptor subunit CD123 is overexpressed in many hematological malignancies, including leukemic stem-like cells of patients with acute myeloid leukemia (AML) [49,50,51,52], progenitor cells of chronic myeloid leukemia (CML) [53], blastic plasmacytoid dendritic cell neoplasm (BPDCN) [54,55], neoplastic mast cells and systemic mastocytosis [56,57]. Of note, CD123 expression in AML correlates with reduced patient survival, supporting the therapeutic targeting of CD123 in this disease [49,50,51,52,54,58,59,60,61,62].

## 3. GM-CSF and IL-3 in Sepsis

Sepsis is defined as a life-threatening organ dysfunction syndrome caused by a dysregulated host response to infection [63]. GM-CSF and IL-3 play a central role in the pathogenesis of sepsis and might also be exploited for diagnostic or therapeutic purposes. In a cecal ligation and puncture (CLP) murine model of sepsis, GM-CSF produced by IRA B cells enhances bacterial clearance and exerts a protective role [40]. Several experimental studies in murine models of sepsis have pointed to a key role for GM-CSF in increasing survival, and therefore GM-CSF has also been tested as an immunostimulating adjuvant therapy in patients with sepsis. Unfortunately and at variance with murine data, clinical results have been inconclusive so far [64]. GM-CSF treatment does not significantly impact 28-day mortality but improves clinical endpoints in selected patient populations. For example, in patients with non-traumatic abdominal sepsis who underwent surgical intervention, GM-CSF treatment significantly decreased the hospital length of stay, duration of antibiotic therapy and infectious complications [65]. Another study found that GM-CSF treatment in patients with severe sepsis and respiratory dysfunction improved gas exchange (which indicates a reduced number of alveolar neutrophils) and increased function of pulmonary phagocytes and circulating neutrophils [66]. Several reasons may account for the failure of GM-CSF treatment to decrease mortality in sepsis patients. First, most of the studies were underpowered to evaluate survival and only assessed 28-day mortality; in addition, GM-CSF treatment might be of limited use across all patient populations but can result beneficial in specific subsets of patients, thereby highlighting the importance of proper patient stratification; finally, timing and modalities of GM-CSF treatment should be harmonized across clinical trials in order to make results more comparable. Since GM-CSF treatment has not been associated with significant adverse effects so far, a large prospective multi-center with standardized GM-CSF administration to sepsis patients with the aim of stratifying patients and disease states is warranted to define a possible role for GM-CSF immunostimulating adjuvant therapy in sepsis management.

The role of IL-3 in sepsis has been less intensively investigated compared to that of GM-CSF. Interestingly, IL-3 has recently been implicated in the pathogenesis of CLP-induced acute sepsis by promoting myelopoiesis and cytokine storm. Genetic deletion or antibody-mediated blockade of IL-3 activity protects mice from sepsis. Of note, high plasma levels of IL-3 in sepsis patients are associated with increased mortality [11]. The recent development of a point-of-care platform for rapid and sensitive detection of IL-3 in human blood samples has further confirmed that increased IL-3 plasma levels are associated with high organ failure rates and holds promises for implementing the use of IL-3 as a biomarker of sepsis in the clinical setting [67].

## 4. GM-CSF and IL-3 in Autoimmune and Allergic Diseases

Several studies have reported a prominent role for GM-CSF and to a lesser extent IL-3 in autoimmune diseases, and for some of them such as multiple sclerosis (MS) [68] and rheumatoid arthritis (RA) [69,70], humanized blocking antibodies are now being evaluated in clinical trials. GM-CSF genetic deletion or blockade impairs the development of, while GM-CSF administration or transgenic expression in T cells exacerbates EAE [26,27,71,72,73]. Of note, GM-CSF expression is a marker of encephalitogenic CD4^+^ T cells in both mice and humans [26,27,29,30,31]. However, GM-CSF expression in MS is not a unique feature of CD4^+^ T cells since it is also produced by memory B cells [28]. Mechanistically, it has been proposed that GM-CSF acts on monocytes to promote the expression of a pathogenic program characterized by genes linked to inflammasome function, phagocytosis and chemotaxis [16]. GM-CSF is currently under investigation as a therapeutic target in patients with MS and a phase Ib clinical trial using a recombinant human anti-GM-CSF antibody has been recently completed [68]. Encephalitogenic CD4^+^ T cells also express IL-3. However, conflicting results exist on the role of this cytokine in the development and progression of EAE [31,74,75].

Similarly to EAE, GM-CSF deficiency or neutralization prevents disease progression in murine models of arthritis, whereas GM-CSF administration leads to symptoms exacerbation [76,77,78,79]. Mechanistically, T cell-derived GM-CSF increases chronic inflammation but is dispensable for arthritis initiation. On the other hand, stromal cells and synovial ILCs represent the main source of GM-CSF [80]. Importantly, GM-CSF levels are increased in synovial fluid of RA patients [81], and a phase IIb clinical trial using a recombinant human anti-GM-CSF antibody in patients with moderate-to-severe RA has shown decreased disease activity [70]. More controversial is the role of IL-3 in arthritis, which has been reported to either attenuate [82] or exacerbate symptoms [83].

IL-3 plays a central role in modulating basophil and mast cell expansion and activity in the context of type 2 (allergic) responses [84,85]. Th2 lymphocytes secrete IL-3 that expand and recruit basophils, which in turn reinforce Th2 polarization [35,36,37]. Basophil-derived cytokines (e.g., IL-4 and IL-13) can also promote monocyte and macrophage alternative activation [15,86]. Importantly, IL-3-deficient mice do not develop airway hyperresponsiveness (AHR) in a model of allergic airway inflammation induced by systemic injections of ovalbumin (OVA) adsorbed onto alum adjuvant and followed by intranasal OVA challenge. Adoptive transfer of IL-3-differentiated basophils restores AHR [19]. GM-CSF also contributes to the development of allergic airway inflammation upon intranasal sensitization with dust mites. In these models, airway epithelial cells release GM-CSF that, together with other cytokines (e.g., IL-33), promotes the activation of lung CD11b^+^ DCs and the ensuing development of Th2 immunity [23,24,25]. Finally, eosinophils (hallmark cell of allergic inflammation) [87] can be activated not only by IL-5 but also by GM-CSF and IL-3 [18,21,88,89]. As such, therapeutic strategies aimed at targeting GM-CSF or βc are currently under investigation for the treatment of asthma [1].

Finally, administration of exogenous IL-3 or an anti-IL-3-blocking antibody to MRL/lpr mice (which develops a spontaneous autoimmune disease that resembles human systemic lupus erythematosus) has shown a prominent role for IL-3 in the pathogenesis and progression of lupus nephritis [90].

## 5. GM-CSF and IL-3 in Cardiovascular Diseases

The immune system is regarded as a major player in the development and progression of cardiovascular diseases [91,92,93,94,95,96,97], including atherosclerosis, myocardial infarction (MI), dilated cardiomyopathies, myocarditis, hypertension, Kawasaki disease (KD), aortic dissection and acute rheumatic fever. In some of these conditions, a prominent role for GM-CSF and/or IL-3 has also been shown. Several lines of evidence support the notion that GM-CSF exerts a pro-atherogenic role. *Ldlr^−/−^ Csf2^−/−^* mice develop smaller lesions [98], whereas exogenous administration of GM-CSF to atherosclerotic mice stimulates intimal cell proliferation and promotes the development of atherosclerosis plaques [99,100]. Interestingly, IRA B cells expand in atherosclerosis and play a pathogenetic role by producing GM-CSF that promotes DC generation and differentiation of IFNγ-secreting Th1 cells [42]. GM-CSF and IL-3 also promote extramedullary hematopoiesis and differentiation of monocytes that infiltrate and aggravate atherosclerotic lesions [101].

Genetic deletion or antibody-mediated blockade of GM-CSF increases survival following MI by reducing inflammation and leukocyte accrual and promoting wound healing. Mechanistically, GM-CSF is produced by cardiac fibroblast shortly after MI, elicits chemokine production in the infarcted tissue (which in turn promotes neutrophil and monocyte recruitment) and also enhances myelopoiesis by acting on a specific βc-expressing progenitor cell population [102]. Collectively, these findings suggest that GM-CSF exerts a detrimental role in the context of MI.

GM-CSF production by cardiac fibroblast has also been implicated in the pathogenesis of KD, a systemic vasculitis of the childhood that particularly affects coronary arteries and therefore represents a leading cause of pediatric heart disease in developed countries [103]. In a mouse model of KD elicited by *Candida albicans* water-soluble fraction (CAWS), GM-CSF is rapidly produced by cardiac fibroblast upon CAWS challenge and stimulates cytokine and chemokine production by resident macrophages, thereby promoting cardiac inflammation. Interestingly, antibody-mediated GM-CSF blockade prevents the development of cardiac inflammation [104].

Group A streptococcus (GAS) infection can lead to a delayed autoimmune response known as acute rheumatic fever (ARF) that can also involve the heart causing autoimmune-drive rheumatic heart disease. Recently, it was reported that peripheral blood mononuclear cells isolated from ARF patients and stimulated in vitro with GAS produce more CD4^+^ T cell-derived GM-CSF compared to healthy controls [105]. It is tempting to speculate that CD4^+^ T cell-derived GM-CSF plays a role in the pathogenesis of ARF. Further studies are required to confirm this hypothesis.

## 6. GM-CSF and IL-3 in Inflammatory Bowel Diseases (IBDs)

IL-23-induced GM-CSF drives the pathogenesis of T cell-driven and ILC-driven experimental models of colitis [18,34,106]. GM-CSF promotes skewing of hematopoietic stem and progenitor cells towards toward granulocyte-monocyte progenitors and also their accumulation in spleen and colon [34]. Interestingly, eosinophils have been identified as a critical effector cell subset of the IL-23/GM-CSF axis in colitis. Mechanistically, GM-CSF promotes eosinophil accumulation in the gut and their activation, which leads to the release of toxic mediators like eosinophil peroxidase and the ensuing tissue damage [18]. Nevertheless, the role of GM-CSF in IBDs is also context-dependent, as DSS-induced epithelial damage is exacerbated in GM-CSF deficient mice [47,48,107], and adoptive transfer of GM-CSF-treated murine monocytes ameliorates T cell-dependent experimental colitis [17].

Several reports in Crohn’s disease (CD) and ulcerative colitis (UC) patients indicate disruption of GM-CSF signaling due to increased circulating or local levels of anti-GM-CSF autoantibodies [108,109,110], downregulation of CD116 [111] or inactivating mutations of βc [112,113]. Although these results point to a protective role for GM-CSF, clinical trials of recombinant GM-CSF administration to CD patients have failed to demonstrate induction of remission [114]. Further studies are required to define the mechanistic role of GM-CSF in IBDs and whether its blockade or administration may be beneficial in specific patient populations.

## 7. GM-CSF and IL-3 in Cancer Therapy

GM-CSF is highly expressed in the milieu of several cancers and has been variably associated with pro- or anti-tumorigenic functions [115,116]. One of the first attempts to use GM-CSF as a therapy for experimental cancer models consisted in treating mice with autologous cancer cells transduced with GM-CSF [117]. Although this approach proved extremely successful in mice, phase 3 clinical trials using GM-CSF-transduced irradiated tumor cells failed to demonstrate clinical efficacy and hampered further development of this approach [118]. However, the recent success of immune checkpoint inhibitors for treatment of hematologic and solid tumors [119,120,121] has sparked renewed interest in the use of GM-CSF as a component of combination therapy. Accordingly, combining GM-CSF to the anti-CTLA-4-blocking antibody Ipilimimab increases the efficacy of the latter in patients with metastatic melanoma [122]. GM-CSF is also expressed by the oncolytic virus talimogene laherparecvec (T-VEC), a modified herpes simplex virus approved for local treatment of non-resectable melanoma [123]. Recently, biomaterial-based vaccination systems encapsulating GM-CSF, TLR agonists (e.g., CpG) and autologous irradiated cancer cells alone or in combination with immune checkpoint inhibitors have shown promising results in experimental cancer models [124,125,126,127,128]. These new approaches are likely to redefine the use of GM-CSF in the context of cancer immunotherapy.

IL-3 is a survival factor for several malignant hematopoietic cell lineages. As such, IL-3 or its specific receptor subunit CD123 has been investigated as therapeutic targets, especially in AML where CD123 expression on cancer cells correlates with reduced patient survival [49,50,51]. Therapeutic strategies tested so far include CD123-directed CAR T cells, IL-3-toxin conjugates and anti-CD123 that can act by blocking IL-3 pro-survival signaling and also promoting NK cell-mediated antibody-dependent cell-mediated cytotoxicity [52,54,58,59,60,61,62].

## 8. GM-CSF and IL-3 in Trained Immunity

A growing body of literature shows that priming of the innate immune system with microbial of inflammatory stimuli can have a long-term impact on the subsequent response to secondary stimuli, a phenomenon referred to as innate immune memory or trained immunity [129,130]. This concept has also been fostered by clinical evidence that live attenuated vaccines exert heterologous (“non-specific”) effects, potentially protecting against unrelated pathogens and having positive effects on survival that are substantially greater than those accounted for by protection against the target disease [131]. In vitro and in vivo experimental models have shown that innate immune cell stimulation with pattern recognition receptor ligands or cytokines can induce trained immunity [14,46,132,133,134,135,136,137,138,139,140,141,142,143,144,145,146]. Recently, an in vivo model of β-glucan-induced trained immunity has shown a remarkable effect on myelopoiesis that was at least in part dependent on GM-CSF activity on bone marrow precursors [143]. GM-CSF also enhances the response to β-glucan in an in vitro experimental model of trained immunity [46]. Interestingly, we have shown that GM-CSF and IL-3 can directly modulate subsequent responses of human monocytes to LPS in in vitro models of trained immunity. In particular, in a short-term model of trained immunity (priming with GM-CSF or IL-3 and LPS stimulation the following day), GM-CSF and IL-3 enhance LPS-induced TNF production at the post-transcriptional level by modulating p38 and nicotinamide adenine dinucleotide (NAD)-dependent sirtuin 2 (SIRT2) activities. Conversely, in a long-term model of trained immunity (priming with GM-CSF or IL-3 followed by 6 days of resting and then LPS stimulation), GM-CSF and IL-3 enhance LPS-induced TNF production by promoting monocyte renewal and increase in cell number in a c-Myc-dependent manner [14]. Altogether, these results point to an important role for GM-CSF (and possibly also IL-3) in trained immunity and also open new ways of understanding its mechanism of action in vitro and in vivo.

## 9. Conclusions

Although the role of GM-CSF and IL-3 in mediating emergency myelopoiesis has been known for a long time, they are now considered critical modulators of the innate immune response by acting directly on mature immune cells. As such, they have been implicated in the pathogenesis of a range of diseases and are now been investigated as immunostimulating agents in cancer therapy (Figure 1). In addition, uncovering their role as mediators of innate immune memory has paved the way for new mechanistic investigations on the effects of GM-CSF and IL-3 on the innate immune system. Despite this growing body of knowledge, several critical questions still remain unanswered. For example, it is unclear to what extent GM-CSF and IL-3 activities overlap and diverge in vitro and in vivo. Likewise, most of the efforts have been devoted to characterizing the effects of GM-CSF, with significantly less examples that have specifically focused the investigation on IL-3. Finally, it is becoming clear that the contribution of GM-CSF and IL-3 to certain diseases (e.g., IBD) can be context-dependent. Therefore, it will be critical to define specific patient populations as well as an appropriate time window in which modulating the activity of these cytokines can lead to clinical benefits.

## Figures and Tables

**Figure 1 ijms-20-00834-f001:**
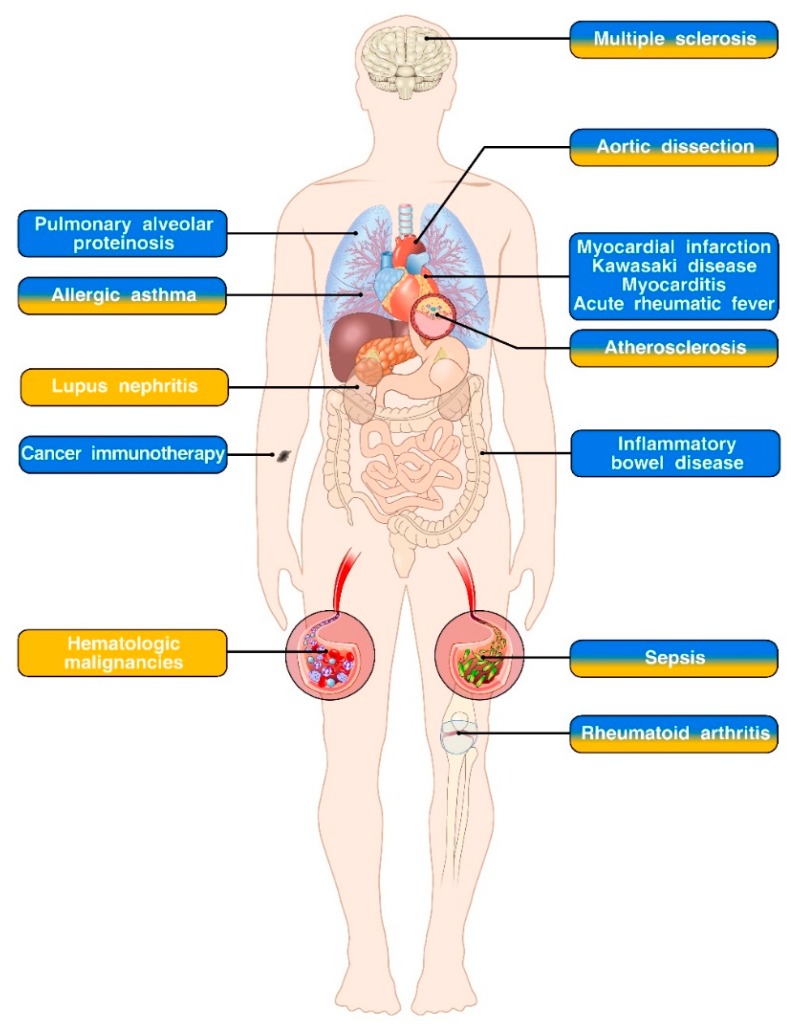
GM-CSF and IL-3 in immune-mediated diseases. Diseases in which a role for GM-CSF, IL-3 or both has been described are respectively indicated in blue, yellow or mixed blue/yellow boxes. Pre-clinical and clinical studies support an important role for GM-CSF in the pathogenesis of pulmonary alveolar proteinosis, cardiovascular diseases (myocardial infarction, Kawasaki disease, myocarditis, and acute rheumatic fever) and inflammatory bowel disease. Interestingly, GM-CSF is an effective immunostimulatory agent in the context of cancer immunotherapy. Both GM-CSF and IL-3 play pivotal roles in the development and progression of allergic asthma, aortic dissection, atherosclerosis and sepsis. While the role of GM-CSF in rheumatoid arthritis and multiple sclerosis has been firmly established, the relevance of IL-3 in the pathogenesis of these disorders is more controversial. There is also evidence of IL-3 involvement in lupus nephritis. Finally, IL-3 acts as a survival factor in several hematologic malignancies and is currently under investigation, together with its specific receptor subunit CD123 as therapeutic target.

**Table 1 ijms-20-00834-t001:** Involvement of GM-CSF and IL-3 in disease pathogenesis.

Disease	Cytokine Involved	Description	References
Pulmonary alveolar proteinosis	GM-CSF	Decreased bioavailability of GM-CSF leads to impaired maturation of alveolar macrophages and accumulation of surfactant and related products within lung alveoli.	[4,6,7,8]
Myocardial infarction	GM-CSF	GM-CSF produced by cardiac fibroblasts promotes inflammation and leukocyte accrual and impairs wound healing upon myocardial infarction.	[102]
Kawasaki disease	GM-CSF	GM-CSF produced by cardiac fibroblasts stimulates cytokine and chemokine production by resident macrophages, thereby promoting cardiac inflammation.	[104]
Myocarditis	GM-CSF	IL-23-induced, CD4^+^ T cell-derived GM-CSF drives cardiac inflammation in an experimental model of myocarditis.	[97]
Acute rheumatic fever	GM-CSF	PBMCs isolated from acute rheumatic fever patients and stimulated in vitro with GAS produce more CD4^+^ T cell-derived GM-CSF compared to healthy controls.	[105]
Inflammatory bowel disease	GM-CSF	The role of GM-CSF in IBD is context dependent. IL-23-induced GM-CSF drives the pathogenesis of T cell-driven and ILC-driven experimental models of colitis. DSS-induced epithelial damage is exacerbated in GM-CSF deficient mice. Adoptive transfer of GM-CSF-treated murine monocytes ameliorates T cell-dependent experimental colitis.	[17,18,34,47,48,106,107]
Lupus nephritis	IL-3	Administration of exogenous IL-3 and an anti-IL-3-blocking antibody to MRL/lpr mice (which develop a spontaneous autoimmune disease that resembles human systemic lupus erythematosus), respectively, promotes and restrains lupus nephritis progression.	[90]
Hematologic malignancies	IL-3	IL-3 is a survival factor for several malignant hematopoietic cell lineages, especially acute myeloid leukemia where CD123 expression on cancer cells correlates with reduced patient survival.	[49,50,51]
Allergic asthma	GM-CSF and IL-3	Epithelial cell-derived GM-CSF promotes DC activation and Th2 immunity in an experimental model of dust mite-elicited allergic airway inflammation. Basophil-derived IL-3 induces the development of AHR in a model of allergic airway inflammation induced by systemic injections of OVA/alum followed by intranasal OVA challenge.	[19,23,24,25]
Multiple sclerosis	GM-CSF and IL-3	GM-CSF promotes the development of EAE by inducing the expression of a pathogenic program in monocytes. GM-CSF can be produced by both CD4^+^ T cells and memory B cells. Encephalitogenic CD4+ T cells also express IL-3. However, conflicting results exist on the role of this cytokine in the development and progression of EAE.	[16,26,27,28,29,30,31,71,72,73,74,75]
Aortic dissection	GM-CSF and IL-3	IL-3 elicits macrophage production of MMP12 and is required for the development of chemically induced thoracic aortic aneurysm and dissection. GM-CSF synergizes with aortic inflammation elicited by CaCl_2_ and Ang II to induce aortic dissection/intramural hematoma.	[95,96]
Atherosclerosis	GM-CSF and IL-3	Both GM-CSF and IL-3 aggravate atherosclerosis development by promoting formation of atherosclerotic plaques, extramedullary hematopoiesis and monocyte differentiation, DC generation and Th1 polarization.	[42,98,99,100,101]
Sepsis	GM-CSF and IL-3	In experimental models of CLP-induced sepsis, GM-CSF produced by IRA B cells enhances bacterial clearance and exerts a protective role. Conversely, IL-3 promotes myelopoiesis and cytokine storm therefore playing a detrimental role.	[11,40]
Rheumatoid arthritis	GM-CSF and IL-3	GM-CSF derived from ILCs, stromal and T cells plays a prominent role in the development and progression of murine models of arthritis. IL-3 has been reported to either attenuate or exacerbate arthritis symptoms.	[76,77,78,79,80,82,83]

Abbreviations: AHR, airway hyperresponsiveness; Ang II, angiotensin II; CLP, cecal ligation and puncture; DC, dendritic cell; DSS, dextran sulfate sodium; EAE, experimental autoimmune encephalomyelitis; GAS, group A streptococcus; IBD, inflammatory bowel disease; ILCs, innate lymphoid cells; IRA B cells, innate response activator B cells; MMP12, matrix metalloprotease 12; OVA, ovalbumin; PBMCs, peripheral blood mononuclear cells.

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
