# Peer review of "Innate Immune Modulation by GM-CSF and IL-3 in Health and Disease"

_ijms, 2019, doi:10.3390/ijms20040834_

Round 1
Reviewer 1 Report
The authors provide a concise review on the role of innate immune cell-derived GM-CSF and IL-3 in different pathophysiological conditions. The article is well written and structured.
However, I think the article would benefit from including the effects of GM-CSF (and IL-3) in tissue regeneration, particularly dermal wound healing as well as during bacterial infections.
Author Response
Point 1: The authors provide a concise review on the role of innate immune cell-derived GM-CSF and IL-3 in different pathophysiological conditions. The article is well written and structured.
Response 1: We thank Reviewer 1 for his/her positive comments to our manuscript.
Point 2:However, I think the article would benefit from including the effects of GM-CSF (and IL-3) in tissue regeneration, particularly dermal wound healing as well as during bacterial infections.
Response 2:We agree with Reviewer 1 that GM-CSF and IL-3 play important roles in tissue regeneration and many other contexts as well. However, we focused our manuscript on those conditions in which GM-CSF and IL-3 exert a direct effect on innate immune cells. Regarding the role of GM-CSF and IL-3 in bacterial infections, we have an entire section dedicated to sepsis, reporting both experimental and clinical evidence. We acknowledge that GM-CSF and IL-3 have relevant roles in several bacterial infections. We decided to focus on sepsis due to experimental and clinical relevance of the topic and availability of clinical data assessing the relevance of GM-CSF and IL-3 as therapeutic targets and/or diagnostic markers in sepsis.
Reviewer 2 Report
- Title: Innate immune modulation by GM-CSF and IL-3 in health and disease.
- In this review, the authors study the role of GM-CSF and IL-3 in innate immunity and conclude that these cytokines can open new therapeutic pathways for different immune-mediated diseases.
- In the abstract they talk about GM-CSF and IL-3 as central mediators of innate immune, and this is correct, but too generic as innate immunity is a very large field. The authors should be more precise.
- The authors should make a clear and brief functional distinction between GM-CSF and IL-3 (it seems that they are similar in this article). Saying, for example, that the former is a more specific growth factor for granulocytes and monocytes, while the latter is a more general upstream growth factor.
- To make the paper more interesting for readers of this journal, and less arid on innate immunity, I also suggest to briefly insert in the discussion of this review, the pro-inflammatory role of cytokines and their potential inhibition with therapeutic effect. Very recently, on these topics, 3 interesting articles have been published, which I suggest to read, incorporate their meaning and report them in the discussion and in the list of references.
- Stimulated mast cells release inflammatory cytokines: potential suppression and therapeutical aspects. Varvara G, Tettamanti L, Gallenga CE, Caraffa A, D'Ovidio C, Mastrangelo F, Ronconi G, Kritas SK, Conti P. J Biol Regul Homeost Agents. 2018 Nov-Dec;32(6):1355-1360.
- Status of cathelicidin IL-37, cytokine TNF, and vitamin D in patients with pulmonary tuberculosis. Majewski K, Agier J, Kozłowska E, Brzezińska-Błaszczyk E. J Biol Regul Homeost Agents. 2018 Mar-Apr;32(2):321-325.
- IL-33 mediates allergy through mast cell activation: Potential inhibitory effect of certain cytokines. Tettamanti L, Kritas SK, Gallenga CE, D'Ovidio C, Mastrangelo F, Ronconi G, Caraffa A, Toniato E, Conti P. J Biol Regul Homeost Agents. 2018 Sep-Oct;32(5):1061-1065.
- A consideration on the authors: they seem to me prepared and experienced and therefore able to write a review
- I suggest accepting this important paper after minor revision.
Author Response
Point 1: In the abstract they talk about GM-CSF and IL-3 as central mediators of innate immune, and this is correct, but too generic as innate immunity is a very large field. The authors should be more precise.
Response 1: We agree with Reviewer 2 that innate immunity is a large field. Indeed, we decided to focus our review on innate immune effector functions elicited only by GM-CSF and IL-3 and their impact in health and disease conditions without extensively discussing the role of GM-CSF and IL-3 in innate immune cell differentiation (i.e. myelopoiesis) or the signaling pathways activated by these cytokines.
Point 2:The authors should make a clear and brief functional distinction between GM-CSF and IL-3 (it seems that they are similar in this article). Saying, for example, that the former is a more specific growth factor for granulocytes and monocytes, while the latter is a more general upstream growth factor.
Response 2:We agree with Reviewer 2 that GM-CSF and IL-3 are functionally distinct. Indeed, throughout the manuscript and in Figure 1 and Table 1 as well we highlight the differential contribution of GM-CSF and IL-3 to disease pathogenesis. We also report which innate immune cell subset is activated by GM-CSF or IL-3 for every disease we mention whenever this information is available in the literature.
Point 3:To make the paper more interesting for readers of this journal, and less arid on innate immunity, I also suggest to briefly insert in the discussion of this review, the pro-inflammatory role of cytokines and their potential inhibition with therapeutic effect. Very recently, on these topics, 3 interesting articles have been published, which I suggest to read, incorporate their meaning and report them in the discussion and in the list of references.
- Stimulated mast cells release inflammatory cytokines: potential suppression and therapeutical aspects. Varvara G, Tettamanti L, Gallenga CE, Caraffa A, D'Ovidio C, Mastrangelo F, Ronconi G, Kritas SK, Conti P. J Biol Regul Homeost Agents. 2018 Nov-Dec;32(6):1355-1360.
- Status of cathelicidin IL-37, cytokine TNF, and vitamin D in patients with pulmonary tuberculosis. Majewski K, Agier J, Kozłowska E, Brzezińska-Błaszczyk E. J Biol Regul Homeost Agents. 2018 Mar-Apr;32(2):321-325.
- IL-33 mediates allergy through mast cell activation: Potential inhibitory effect of certain cytokines. Tettamanti L, Kritas SK, Gallenga CE, D'Ovidio C, Mastrangelo F, Ronconi G, Caraffa A, Toniato E, Conti P. J Biol Regul Homeost Agents. 2018 Sep-Oct;32(5):1061-1065.
Response 3:We acknowledge the importance of the articles suggested by Reviewer 2. Although we agree in principle that a broader discussion on pro-inflammatory cytokines, their contribution to disease pathogenesis and their potential as therapeutic targets might be of interest, we also believe that it would be important to keep the focus of our review on GM-CSF and IL-3 since these cytokines have recently been identified as potent modulators of innate immune effector functions and in order to avoid a lengthy and vague discussion of the literature.
Point 4: A consideration on the authors: they seem to me prepared and experienced and therefore able to write a review. I suggest accepting this important paper after minor revision.
Response 4: We thank Reviewer 2 for his/her positive comments to our work.
Round 2
Reviewer 1 Report
Given the strict focus of the review article, the authors refrain from the recommended changes. Therefore, I would like to leave it to the editor whether this is appropriate.